# Mechanocaloric effects in superionic thin films from atomistic simulations

Arun K. Sagotra[1,2], Daniel Errandonea [iD] [3] & Claudio Cazorla[1,2]

Solid-state cooling is an energy-efficient and scalable refrigeration technology that exploits the adiabatic variation of a crystalline order parameter under an external field (electric, magnetic, or mechanic). The mechanocaloric effect bears one of the greatest cooling potentials in terms of energy efficiency owing to its large available latent heat. Here we show that giant mechanocaloric effects occur in thin films of well-known families of fast-ion conductors, namely Li-rich ($Li_3OCl$) and type-I (AgI), an abundant class of materials that routinely are employed in electrochemistry cells. Our simulations reveal that at room temperature AgI undergoes an adiabatic temperature shift of 38 K under a biaxial stress of 1 GPa. Likewise, $Li_3OCl$ displays a cooling capacity of 9 K under similar mechanical conditions although at a considerably higher temperature. We also show that ionic vacancies have a detrimental effect on the cooling performance of superionic thin films. Our findings should motivate experimental mechanocaloric searches in a wide variety of already known superionic materials.

[1] School of Materials Science and Engineering, UNSW Australia, Sydney, NSW 2052, Australia. [2] Integrated Materials Design Centre, UNSW Australia, Sydney, NSW 2052, Australia. [3] Departamento de Física Aplicada (ICMUV), Malta Consolider Team, Universitat de Valencia, 46100 Burjassot, Spain. Correspondence and requests for materials should be addressed to C.C. (email: c.cazorla@unsw.edu.au)

Conventional cooling methods based on gas-compression cycles present a series of critical drawbacks including the use of environmental hazards and lack of scalability to micro sizes. Solid-state cooling represents an elegant solution to all these issues, with mechanocaloric (MC) effects possibly holding the greatest promise in terms of energy efficiency[1]. Superelastic shape-memory alloys (SMA) displaying first-order martensitic transformations between a high-$T$ martensitic and a low-$T$ austenitic phase (e.g., Ni-Ti, Cu-Al-Ni, and Cu-Zn-Sn alloys), are archetypal MC compounds[2]. For example, a giant adiabatic temperature shift of 25.5 K has been measured in NiTi wires upon a small tensile stress of 0.65 GPa[3]. Nevertheless, there are still several hurdles that need to be overcome in order to develop successful MC commercial applications. For instance, the first-order nature of the martensitic transformation involves concomitant hysteresis losses, which is detrimental for cooling efficiency[4], and as the size of SMA shrinks towards the nanoscale the martensitic phase transformation may be suppressed due to overstabilisation of the high-$T$ distorted phase[5]. Finding novel MC materials with sharp second-order phase transitions occurring near room temperature and which persist down to the nanoscale, therefore, may advance the field of solid-state cooling.

Ferroelectric compounds (FE) like, for instance, BaTiO$_3$[6], PbTiO$_3$[7], and Ba$_{1-x}$Ca$_x$Ti$_{1-y}$Zr$_y$O$_3$ solid solutions[8] (generally exploited in sensing, information storage and energy applications), can be synthesised as nano-sized materials and typically exhibit displacive second-order phase transitions[9]. However, current adiabatic temperature shifts, $|\Delta T|$, estimated in most FE materials near room temperature are about one order of magnitude smaller than those achieved in SMA (made the exception of the ferrielectric compound (NH$_4$)$_2$SO$_4$[10]), and the involved mechanical stresses appear to be unsuitably too large ($|\sigma| \gg 1$ GPa). Consequently, the MC strengths reported so far for FE are rather poor in general, namely, $|\Delta T|/|\sigma| < 1$ K·GPa$^{-1}$ (again, made the exception of (NH$_4$)$_2$SO$_4$[10]). Recently, a giant MC effect has been predicted in fluorite-structured fast-ion conductors (FIC), typified by CaF$_2$ and PbF$_2$, which is comparable in magnitude to the benchmark adiabatic temperature shifts measured in SMA[11]. The MC effect disclosed in FIC may be understood in terms of stress-driven changes in ionic diffusivity, which in turn cause large variations in the entropy and dimensions of the material[11]. The originating superionic transition is of second-order type, in analogy to archetypal FE, and fluorite-structured FIC also can be synthesised as nanomaterials[12]. Unfortunately, the superionic transition temperatures in fluorite-structured FIC are far above ambient conditions (i.e., $T_s = 1350$ in CaF$_2$ and 700 K in PbF$_2$[11, 13]), thus hindering the development of likely solid-state cooling applications.

Here we demonstrate giant MC effects in other predominant families of FIC, namely Li-rich (Li$_3$OCl) and type-I (AgI) compounds (following Hull's notation[14]), some of them at room temperature. We use atomistic computer simulations based on force fields and density functional theory to estimate the isothermal entropy and adiabatic temperature shifts attained in FIC with biaxial stresses ($\sigma_{xx} = \sigma_{yy}$ and $\sigma_{zz} = 0$). (Biaxial stresses, either compressive, $\sigma > 0$, or tensile, $\sigma < 0$, are realisable in thin films[15]; dynamic biaxial stresses leading to epitaxial strain changes of the order of 0.1–1% have been demonstrated in thin films by using ferroelectric substrates[16, 17], and larger dynamic changes can be envisaged by means of mechanical extensometer techniques applied on flexible polymeric substrates[18] and of nanoindentation methods[19].) In AgI thin films, we find a room-temperature adiabatic temperature shift of $|\Delta T| = 38$ K for a moderate compressive load of $|\sigma| = 1$ GPa. This result exceeds the corresponding values estimated in FE at $T = 300$ K ($\sim 1$ K[6]) and equals in magnitude the MC records set in

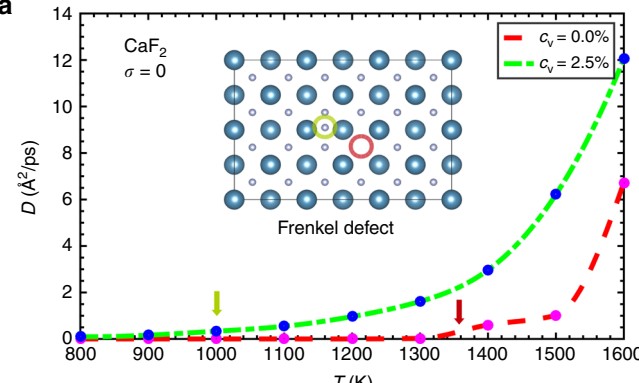

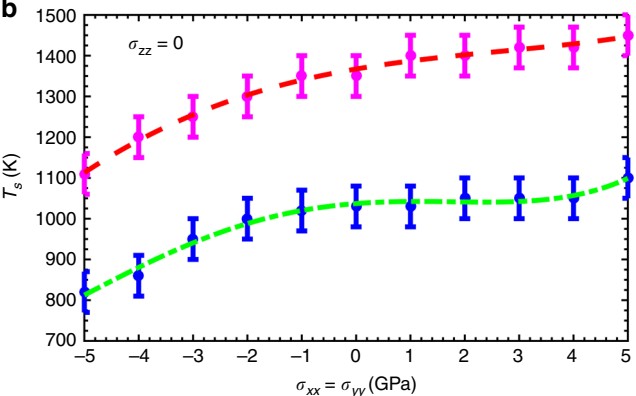

**Fig. 1** Effects of vacancies on the ionic conductivity of type-II thin films. **a** F$^-$ diffusion coefficient in perfect and defective ($c_v = 2.5\%$) calcium fluoride expressed as a function of temperature. The creation of a Frenkel pair defect, the fundamental atomistic mechanism for superionic transport in type-II FIC, is sketched. Red and green arrows indicate the critical superionic transition in the perfect and defective systems. **b** Critical superionic temperature expressed as a function of biaxial stress in perfect and defective CaF$_2$. The lines are guides to the eye and the *error bars* in **b** correspond to the resolution of our calculations

SMA ($\sim 10$ K[3, 20]). Our findings in Li$_3$OCl thin films, namely $|\Delta T| = 9$ K for $|\sigma| = 1$ GPa at $T = 1000$ K, suggest that analogous Li-rich FIC with lower superionic transition temperatures (e.g., Li$_{10}$GeP$_2$S$_{12}$[21]) should display giant MC effects as well. Therefore, we argue that solid-state cooling could benefit immensely from the intensive research already undertaken on solid-state electrochemical batteries[22].

## Results

**Effects of vacancies on the mechanocaloric performance of thin films**. Vacancies are known to enhance significantly ionic transport and lower the superionic critical temperature in FIC[23, 24]. This is explicitly shown in Fig. 1a, where we plot the F$^-$ diffusion coefficient, $D$ (see Methods section), calculated for unstrained CaF$_2$ (type-II FIC[14]) in the absence and presence of ionic vacancies as a function of temperature (note that $D$ increases by >80% in the system containing vacancies). In Fig. 1b, we compare the critical superionic temperature obtained in perfect and defective thin films as a function of biaxial stress; in the $c_v = 2.5\%$ case, $T_s$ is reduced by ~300 K almost independently of $\sigma$. This outcome can be rationalised in terms of a steady lowering of the energy barrier and creation energy of Frenkel pair defects (see Fig. 1a), which is due to an increase of the space available to interstitial ions. Therefore, intuitively one might expect that by introducing ionic vacancies the mechanocaloric

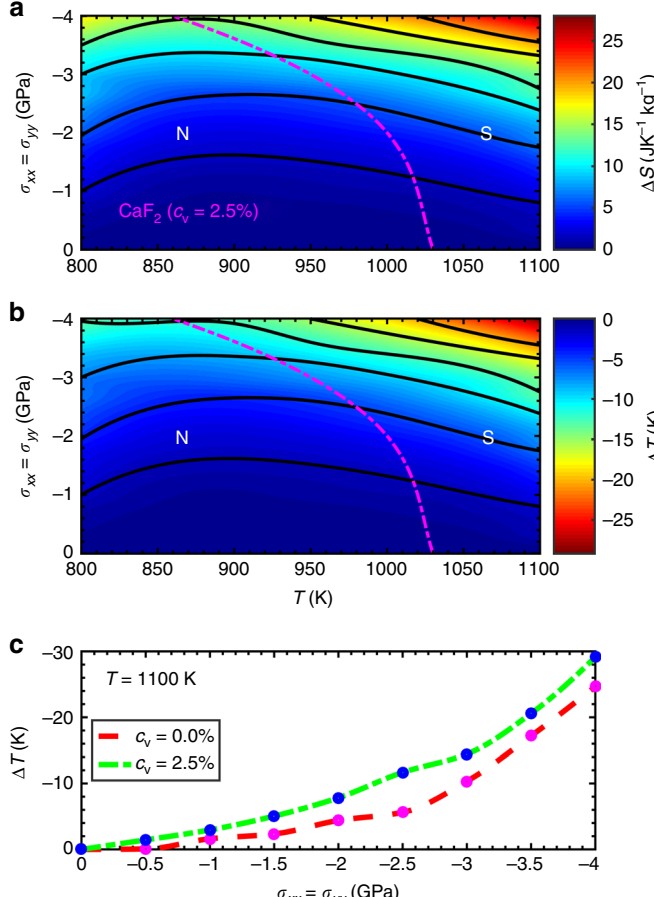

**Fig. 2** Effects of vacancies on the mechanocaloric performance of type-II superionic thin films. Isothermal entropy **a** and adiabatic temperature **b** shifts calculated in defective $CaF_2$ ($c_v = 2.5\%$) as a function of temperature and applied biaxial tensile stress. N and S denote normal and superionic states and the magenta dashed line depicts the boundary between them. **c** Comparison of the adiabatic temperature shifts calculated in perfect and defective $CaF_2$ thin films as a function of biaxial stress at a fixed temperature of 1100 K. Lines in **c** are guides to the eye

performance of FIC should be enhanced. We have found, however, that this is not actually the case in archetypal FIC $CaF_2$ (see Supplementary Figs. 1 and 2).

In Fig. 2a, b, we show the isothermal entropy and adiabatic temperature changes, $\Delta S$ and $\Delta T$ (see Methods section), calculated in type-II FIC thin films with an arbitrary $F^-$ vacancy concentration of 2.5%. Large values of 30 $JK^{-1}$ $kg^{-1}$ and −30 K are obtained respectively at the highest temperature and tensile stress ($T = 1100$ K and $|\sigma| = 4$ GPa), conditions at which the defective system is fully superionic. Upon comparison of the $\Delta T$ values obtained in perfect and defective $CaF_2$ thin films at a fixed temperature of 1100 K (see Fig. 2c), however, we realise that the presence of anion vacancies does not produce any significant enhancement in MC performance (it is worth noticing that at the imposed conditions the perfect system remains in the normal state). Actually, the same applied maximum stress leads to $|\Delta T|$ values two times larger in thin films without vacancies for a slightly higher temperature at which the perfect system becomes superionic[11]. The main cause for this outcome is that the presence of ionic vacancies makes the superionic transition to be less abrupt, by smoothing the concomitant lattice strain associated to the prompt increase of ionic diffusivity (see Supplementary Methods and Supplementary Fig. 5). An analogous effect

is observed also on the anion diffusion coefficient, which in the $c_v = 2.5\%$ case displays a steady, rather than a sudden, increase near the superionic transition point (see Fig. 1a). Therefore, we may conclude that despite ionic vacancies in general favor ionic conductivity their effects on the MC performance of FIC may be adverse.

**Mechanocaloric effect in Li-rich superionic thin films.** We choose $Li_3OCl$ as a representative member of the family of Li-rich FIC[25]. This compound adopts an antiperovskite phase characterised by Li, Cl, and O atoms placed at the octahedral vertices, octahedral centers, and center of a cubic unit cell, respectively (space group $Fm\overline{3}m$, see Supplementary Figs. 1 and 3). We note that $Li_3OCl$ has already been synthesised and characterised in thin-film geometry[26, 27]. The fundamental ion-migration mechanism in this material is related to the presence of vacancies: in the absence of points defects the diffusivity of $Li^+$ ions is null below the corresponding melting point[23, 24]. Meanwhile, the accompanying superionic temperature, $T_s$, strongly depends on the concentration of ionic vacancies (see Supplementary Discussion and Supplementary Fig. 6). Here we analyse the $c_v = 2.5\%$ case, which according to our molecular dynamics simulations (see Methods section) renders a transition temperature of $T_s = 1000$ K at $\sigma = 0$ conditions. In what follows, we focus on tensile stresses ($\sigma < 0$) as in $Li_3OCl$ these favor ionic conductivity the most.

In Fig. 3a, b, we show the $Li^+$ diffusion coefficient and in-plane strain, $\epsilon$ (see Methods section), calculated in $Li_3OCl$ as a function of negative biaxial stress at a fixed temperature of 1000 K. Both quantities increase under applied tensile stress, and in the case of $\epsilon$ the stress-induced enhancement is almost linear. As the ionic diffusivity in the thin film increases so does the isothermal entropy change (Fig. 3c), yielding a value of $\Delta S = 16$ $J \cdot K^{-1}$ $kg^{-1}$ at the maximum tensile stress $\sigma_{max} = −1$ GPa. The accompanying adiabatic temperature change is $|\Delta T| = 9$ K (Fig. 3d), which is about two times larger than the one calculated in $CaF_2$ thin films considering identical ($\sigma$, $T$) conditions and vacancy concentration (see Fig. 2b). The main reason behind this difference is the larger isothermal entropy change found in $Li_3OCl$, which fundamentally is related to the way in which ions hop through favorable pathways within the corresponding structural frameworks.

The largest contribution to the entropy of the superionic transformation can be assumed to be due to the configurational degrees of freedom[28]. Consequently, it is reasonable to expect finding comparable isothermal entropy changes to those attributed to $Li_3OCl$ in other related Li-rich FIC with transition temperatures close to ambient (e.g., $Li_{10}GeP_2S_{12}$[21]). Actually, larger $|\Delta T|$ values may be anticipated in those cases as the cooling efficiency of crystals is enhanced at low temperatures (i.e., the corresponding heat capacity, $C_\sigma$, is small and depends on temperature as $\propto T^3$, see Methods section).

**Mechanocaloric effect in type-I superionic thin films.** We select AgI as a representative member of the family of type-I FIC. At ambient conditions bulk AgI is found as a mixture of wurtzite ($\beta$, hexagonal $P6_3mc$) and zincblende ($\gamma$, cubic $F\overline{4}3m$) phases[14]. As temperature is increased beyond $T_s \sim 400$ K, bulk AgI transforms into a superionic phase ($\alpha$) in which the $I^-$ anions arrange in a cubic bcc lattice and the $Ag^+$ cations are mobile. In practice, AgI thin films can be synthesised either in the $\beta$ or $\gamma$ phase depending on the employed preparation method[29, 30]. For reasons that will become clear later on, we investigate here the cubic $\gamma$ phase consisting of two interlaced monatomic fcc sublattices with fourfold ionic coordination (see Supplementary Figs. 1 and 4). Our following analysis is restricted to compressive

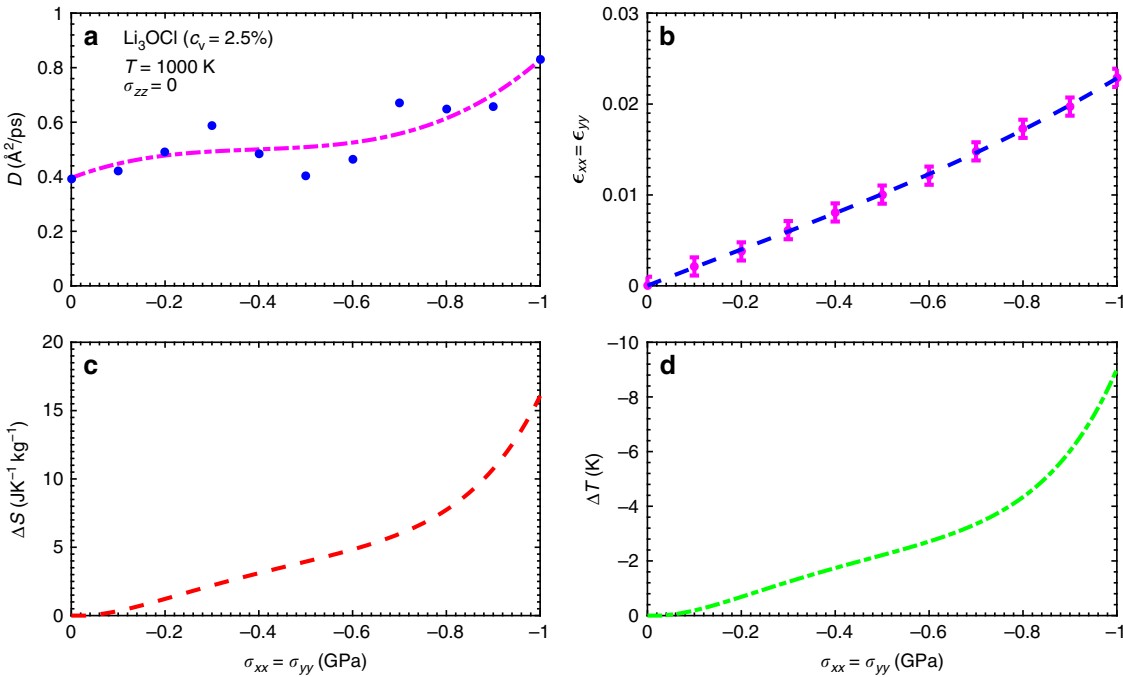

**Fig. 3** Mechanocaloric effect in the Li₃OCl superionic conductor with a point-defect concentration of 2.5%. The Li$^+$ diffusion coefficient **a**, in-plane strain **b**, isothermal entropy shift **c**, and adiabatic temperature change **d** estimated at $T = 1000$ K and expressed as a function of biaxial (tensile) stress. Lines in **a**, **b** are guides to the eye, and the *error bars* in **b** correspond to the standard deviation from 8,000 configurations generated during the simulations

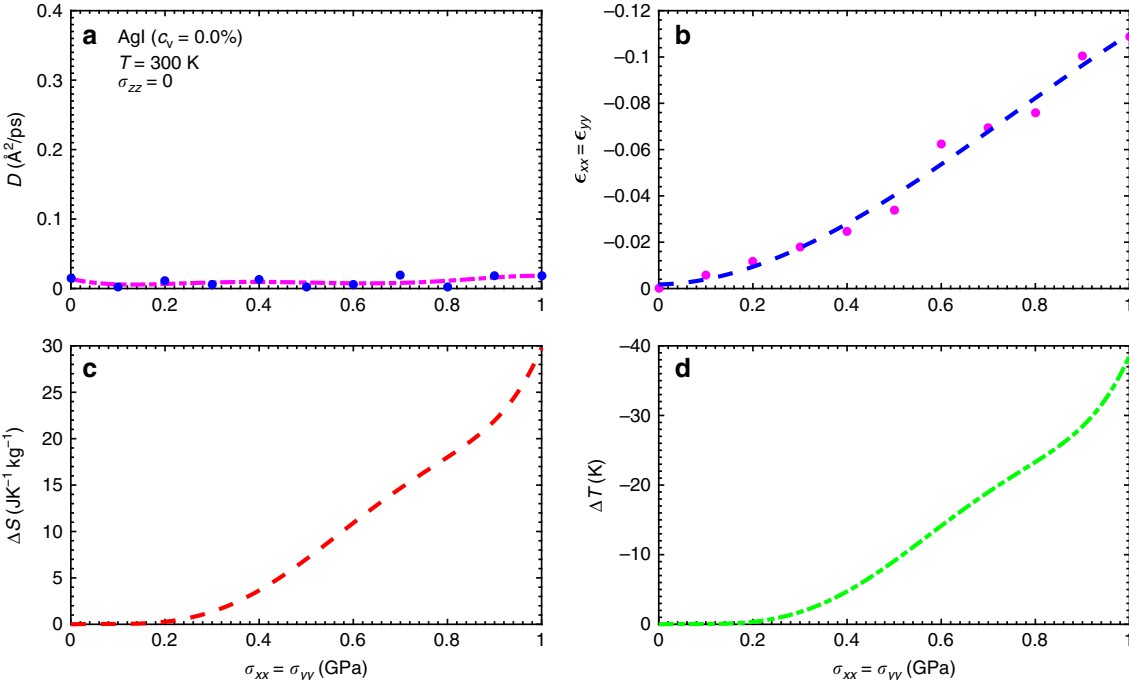

**Fig. 4** Mechanocaloric effect in the AgI superionic conductor without vacancies at $T = 300$ K. The Ag$^+$ diffusion coefficient **a**, in-plane strain **b**, isothermal entropy shift **c**, and adiabatic temperature change **d** estimated at room temperature and expressed as a function of biaxial (compressive) stress. Lines in **a**, **b** are guides to the

stresses ($\sigma > 0$) as we find the largest MC effect at such conditions.

In Figs. 4a and 5a, we show the Ag$^+$ diffusion coefficient calculated in non-defective AgI thin films as a function of compressive biaxial stress at $T = 300$ and 400 K, respectively. It is appreciated that only at the highest analysed temperature and for

stresses larger than 0.8 GPa the system becomes fully superionic (i.e., $D \gg 0$). Nevertheless, the isothermal entropy and adiabatic temperature changes estimated for the maximum stress $\sigma_{max} = 1$ GPa are giant in both cases (see panels c, d in Figs. 4 and 5). In particular, we obtain $\Delta S = 30$ J·K$^{-1}$ kg$^{-1}$ and $\Delta T = 38$ K at $T = 300$ K, and 35 J·K$^{-1}$ kg$^{-1}$ and 51 K at $T = 400$ K. We note

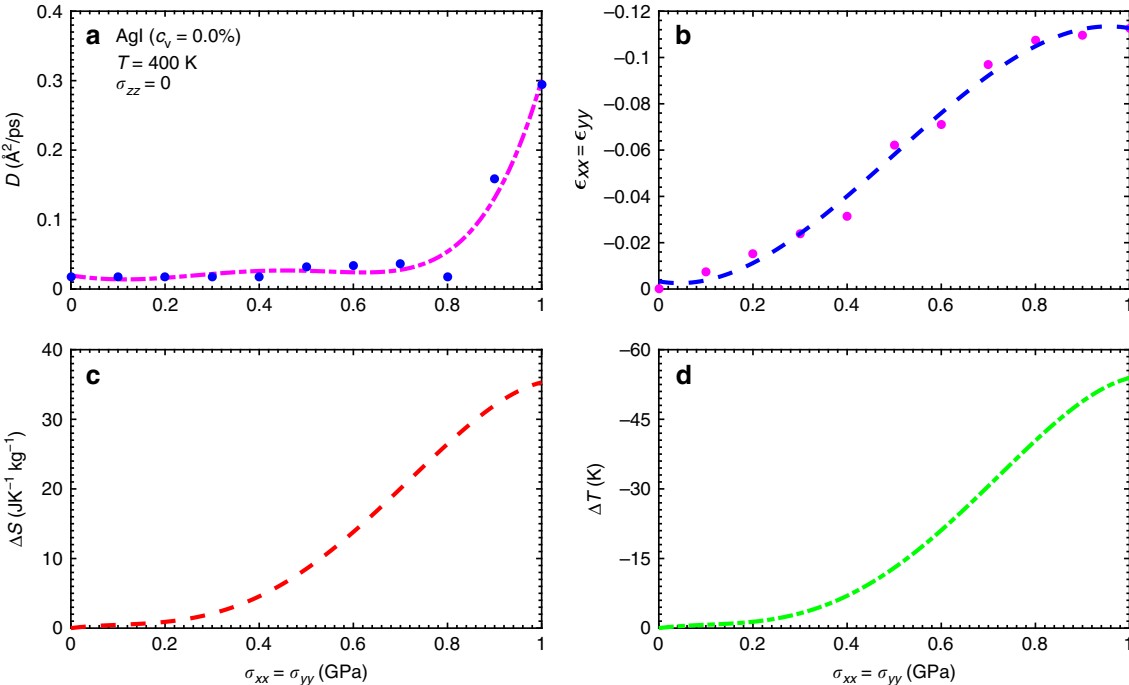

**Fig. 5** Mechanocaloric effect in the AgI superionic conductor without vacancies at $T = 400$ K. The Ag$^+$ diffusion coefficient **a**, in-plane strain **b**, isothermal entropy shift **c**, and adiabatic temperature change **d** estimated at high temperature and expressed as a function of biaxial (compressive) stress. Lines in **a**, **b** are guides to the eye

that the in-plane strains calculated in both cases are also considerably large; as compared to Li$_3$OCl thin films, for instance, those are about four times larger in absolute value (see Figs. 3b–5b). As $T$ is increased beyond room temperature, the calculated entropy and temperature shifts become larger essentially due to the enhanced mobility of Ag$^+$ cations (see Fig. 5a).

At room temperature, the MC performance of AgI thin films commences to be appreciable and to increase steadily for stresses larger than $\sigma_c = 0.2$ GPa (see Fig. 4c, d). This finding signals the triggering of a structural phase transformation at around $\sigma_c$ different from the superionic transition, which occurs at higher temperatures. The continuous variation of the in-plane strain and cation diffusion coefficient as driven by compressive stress indicate that this phase transition is of second-order type. To get microscopic insight into such a structural transformation, we analyse the coordination number, radial pair distribution function, mean squared displacement, and density distribution of I$^-$ and Ag$^+$ ions under different temperature and stress conditions (see Methods section and Supplementary Figs. 7–9).

Our simulation results reveal the existence of a $\sigma$-induced diffusionless order-disorder phase transition involving sizeable displacements of the ionic equilibrium positions with respect to the original zincblende structure. Specifically, both I-I and Ag-Ag coordination numbers amount to 12 in average similarly to what is found in the two monoatomic fcc lattices of reference. However, a precise determination of neighbouring atomic shells from the corresponding radial pair distribution functions, $g(r)$, is not possible for distances larger than few angstroms at biaxial stresses higher than $\sigma_c$ (see Fig. 6). Furthermore, at room temperature and $\sigma_c < \sigma$ conditions the asymptotic behavior of the ionic pair distribution functions, namely, $g(r) \approx 1$, is reached very rapidly with the radial distance ($r \sim 10$ Å, see Fig. 6a, b); this outcome evidences lack of solid translational invariance, in analogy to what is observed in glassy systems. We assuredly identify these features, and others shown in the Supplementary

Figs. 7–9 (e.g., ionic mean squared displacement and density distribution plots), with the presence of atomic disorder in the I$^-$ and Ag$^+$ sublattices at $\sigma_c < \sigma$ conditions. It is worth noticing that, as we have explicitly checked, neither the hexagonal wurtzite nor the cubic rock-salt structures found in bulk AgI at ambient and high-pressure conditions[14] transform to a disordered phase when applying biaxial compressive stresses of ~1 GPa to them at room temperature (see Supplementary Fig. 10).

**Discussion**
The giant room-temperature MC effect revealed in AgI thin films, which is originated by a $\sigma$-induced diffusionless order-disorder phase transition, appears to be very promising in terms of maximum adiabatic temperature shift and mechanocaloric strength, $|\Delta T|/|\sigma|$. In Table 1, we compare the cooling properties of this FIC with those of other well-established MC compounds that have been reported at ambient and near ambient conditions. First, we note that the adiabatic temperature shift estimated in AgI ($|\Delta T| = 38$ K) is equivalent in magnitude to the MC benchmarks obtained in archetypal shape-memory alloys like, for instance, NiTi (e.g., $|\Delta T| = 25.5$ K[3]). As compared to ferroelectric materials, the estimated $\Delta T$ is several times larger in absolute value. In terms of mechanocaloric strength, AgI also shows great promise as it follows closely to shape-memory alloys ($|\Delta T|/|\sigma| \sim 10$ K GPa$^{-1}$[20]) and outperforms perovskite oxide materials (e.g., BaTiO$_3$ with $|\Delta T|/|\sigma| \sim 1$ K GPa$^{-1}$[6]).

Our findings should stimulate the development of new cooling devices based on FIC whose energy efficiency as compared to magnetocaloric and electrocaloric materials is very auspicious. In contrast to other mechanocaloric materials driven by first-order transitions, mechanical hysteresis losses and scalability limitations towards nanosizes should be absent in FIC. Mechanical stresses other than biaxial (i.e., uniaxial and hydrostatic) also can be expected to produce similar mechanocaloric responses than reported here in FIC[11]. In this context, the rich variety of

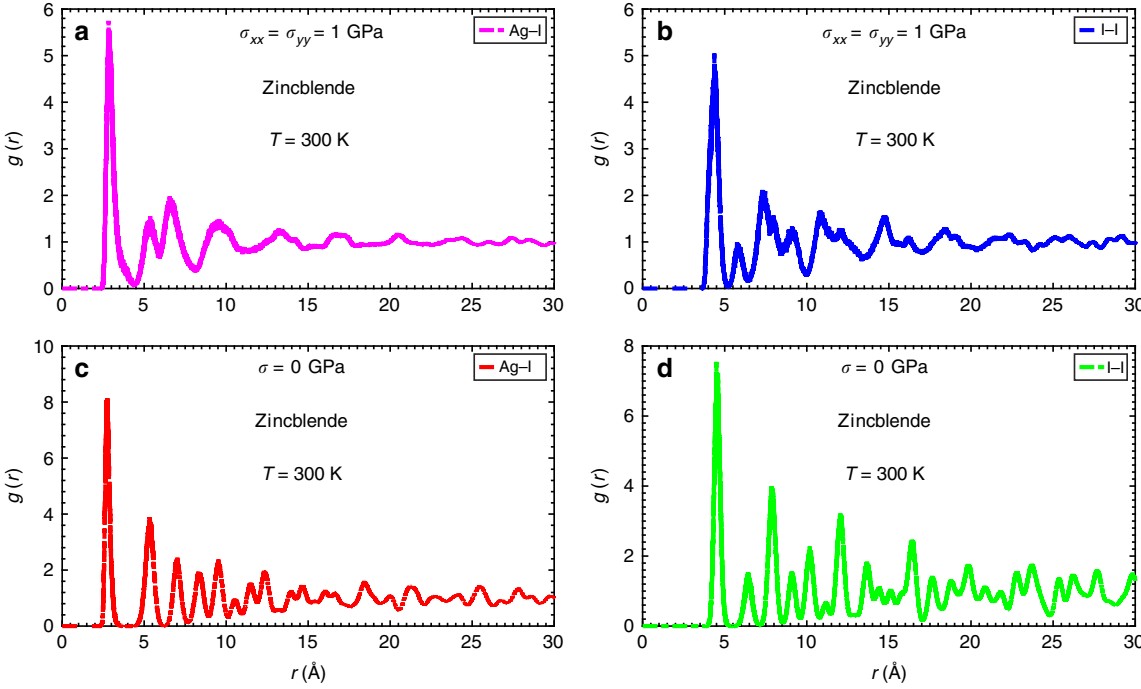

**Fig. 6** Ionic radial pair distribution functions in AgI thin films with the zincblende ($\gamma$) structure at $T = 300$ K. Results are expressed as a function of ionic pairs and biaxial compressive stress. **a** Ag-I and $\sigma_{xx} = \sigma_{yy} = +1$ GPa; **b** I-I and $\sigma_{xx} = \sigma_{yy} = +1$ GPa; **c** Ag-I and $\sigma_{xx} = \sigma_{yy} = 0$ GPa; **d** I-I and $\sigma_{xx} = \sigma_{yy} = 0$ GPa

### Table 1 Giant mechanocaloric (MC) effects near room temperature

| Giant MC material | $T$ (K) | $|\sigma|$ (GPa) | $|\Delta S|$ (JK$^{-1}$ Kg$^{-1}$) | $|\Delta T|$ (K) | $|\Delta T|/|\sigma|$ (K GPa$^{-1}$) | Material type | Ref. |
|---|---|---|---|---|---|---|---|
| NiTi | 295 | 0.65 | — | 25.5 | 39.3 | SMA | 3 |
| Ni$_{49.26}$Mn$_{36.08}$In$_{14.66}$ | 293 | 0.26 | 24.0 | 4.5 | 17.3 | SMA | 20 |
| Fe$_{49}$Rh$_{51}$ | 308 | 0.11 | 12.5 | 8.1 | 73.6 | SMA | 35 |
| Cu$_{68}$Zn$_{16}$Al$_{16}$ | 300 | 0.28 | 16.0 | 15.0 | 53.6 | SMA | 36 |
| BaTiO$_3$ | 300 | 6.50 | 8.0 | 5.5 | 0.9 | FE | 6 |
| (NH$_4$)$_2$SO$_4$ | 219 | 0.10 | 60.0 | 8.0 | 80.0 | FE | 10 |
| AgI | 300 | 1.00 | 30.0 | 38.0 | 38.0 | FIC | This work |

$T$ represents working temperature, $|\sigma|$ applied mechanical stress, $|\Delta S|$ isothermal entropy change, $|\Delta T|$ adiabatic temperature change, $|\Delta T|/|\sigma|$ mechanocaloric strength, SMA shape-memory alloy, FE ferroelectric, and FIC fast-ion conductor. $|\Delta T|$ values have been obtained by using zero-pressure specific heat capacities

superionic materials that have been already investigated with a focus on electrochemical applications grants solid-state cooling with vast new possibilities.

## Methods

**Classical molecular dynamics simulations**. Molecular dynamics ($N$, $P$, $T$) simulations are performed with the LAMMPS code[31]. The pressure and temperature in the system are kept fluctuating around a set-point value by using thermostatting and barostatting techniques in which some dynamic variables are coupled to the particle velocities and simulation box dimensions. The interactions between atoms are modeled with rigid-ion Born-Mayer-Huggins potentials. Large simulation boxes, typically containing 6,144 atoms, are used in which periodic boundary conditions along the three Cartesian directions are applied. Newton's equations of motion are integrated using the customary Verlet's algorithm with a time-step length of $10^{-3}$ ps. A particle-particle particle-mesh $k$-space solver is used to compute long-range van der Waals and Coulomb interactions and forces beyond a cut-off distance of 12 Å at each time step.

We note that by using periodic boundary conditions in our calculations we avoid to explicitly simulate the substrate over which the thin film is grown in practice. Also, possible elastic relaxation effects and the interactions of the thin film with the vacuum at the top surface are totally neglected. Consequently, the simulations are performed very efficiently in terms of computational expense and the fundamental mechanocaloric effects occurring in FIC thin films can be singled out. Further details of our classical molecular dynamics simulations (e.g., interatomic potential models) can be found in the Supplementary Methods and Supplementary Tables 1 and 2.

**Density functional theory calculations**. First-principles DFT calculations are performed to analyse the energy and structural properties of CaF$_2$, Li$_3$OCl, and AgI thin films at zero temperature (see Supplementary Figs. 2–4). We perform these calculations with the VASP code[32] by following the generalized gradient approximation for the exchange-correlation energy due to Perdew et al.[33]. The projector augmented-wave method is used to represent the ionic cores[34], and the electronic states 2s-3s-3p-4s of Ca, 2s-2p of F, 1s-2s of Li, 2s-2p of O, 2s-2p of Cl, 4d-5s of Ag, and 5s-5p of I, are considered as valence. Wave functions are represented in a plane-wave basis truncated at 650 eV. By using these parameters and dense **k**-point grids for Brillouin zone integration, the resulting energies are converged to within 1 meV per formula unit. In the geometry relaxations, a tolerance of 0.01 eV·Å$^{-1}$ is imposed in the atomic forces. We also perform *ab initio* molecular dynamics calculations in order to validate the reliability of the interatomic potential models employed in the classical molecular dynamics simulations. Details of these calculations can be found in the Supplementary Discussion.

**Estimation of key quantities**. The ionic diffusion coefficients are determined as:

$$D = \lim_{t \to \infty} \frac{\langle |R_i(t + t_0) - R_i(t_0)|^2 \rangle}{6t}, \tag{1}$$

where $R_i(t)$ is the position of the migrating ion labelled as $i$ at time $t$, $t_0$ an arbitrary time origin, and $\langle \cdots \rangle$ denotes average over time and particles. Meanwhile, the mean squared displacement of each ionic species is defined as $\langle \Delta R_i^2(t) \rangle \equiv \langle |R_i(t + t_0) - R_i(t_0)|^2 \rangle$.

Owing to the cubic symmetry of the thin films considered in this study, strains $\epsilon_{xx}$ and $\epsilon_{yy}$ and stresses $\sigma_{xx}$ and $\sigma_{yy}$ are identical. Consequently, the accompanying

isothermal entropy changes can be estimated with the formula:

$$\Delta S(\sigma_f, T) = V_0 \cdot \int_0^{|\sigma_f|} \left(\frac{\partial \epsilon_{xx}}{\partial T}\right)_\sigma d\sigma_{xx} + \left(\frac{\partial \epsilon_{yy}}{\partial T}\right)_\sigma d\sigma_{yy}, \quad (2)$$

where $V_0(T) \equiv L_{x,0}(T) \cdot L_{y,0}(T) \cdot L_{z,0}(T)$ is the $T$-dependent volume of the crystal at equilibrium (i.e., considering $\sigma = 0$ conditions), $L_i$ represents the length of the simulation box along the $i$ Cartesian direction, and the mechanical strain components are defined as $\epsilon_{ii}(\sigma, T) \equiv \frac{L_i(\sigma, T) - L_{i,0}(T)}{L_{i,0}(T)}$. Regarding the adiabatic temperature shifts, those are calculated as:

$$\Delta T(\sigma_f, T) = -\int_0^{|\sigma_f|} \frac{T}{C_\sigma(\sigma, T)} \cdot dS, \quad (3)$$

where $C_\sigma(\sigma, T)$ is the heat capacity of the crystal calculated at fixed $\sigma$. In this study we assumed that $C_\sigma(\sigma, T) \approx C_\sigma(0, T)$. Further technical details on our calculations can be found in the Supplementary Methods.

**Data availability**. The data that support the findings of this study are available from the corresponding author (C.C.) upon reasonable request.

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

## Acknowledgements

This research was supported under the Australian Research Council's Future Fellowship funding scheme (No. FT140100135). Computational resources and technical assistance were provided by the Australian Government and the Government of Western Australia through Magnus under the National Computational Merit Allocation Scheme and The Pawsey Supercomputing Centre. D.E. acknowledges financial support from Spanish government MINECO, the Spanish Agencia Estatal de Investigacion (AEI), and Fondo Europeo de Desarrollo Regional (FEDER) under Grants No. MAT2016-75586-C4-1-P and MAT2015-71070-REDC.

## Author contributions

C.C. conceived the study and planned the research. C.C. and A.K.S. performed the theoretical calculations. Results were discussed by C.C., A.K.S. and D.E. The manuscript was written by C.C. and A.K.S. with substantial input from D.E.

## Additional information

**Competing interests:** The authors declare no competing financial interests.

