## [Peer Review File · Nature Communications]

Reviewers' Comments:

Reviewer #1:

Remarks to the Author:

The paper reports predictions of mechanocaloric effects in ionic conductors (38 K for 1 GPa in AgI near room temperature, and 9 K for 1 GPa in Li₃OCl at high temperature). The main result differs from the authors' previous work [9] because the effects in AgI occur near room temperature. Compared with other materials in Table 1, the predicted temperature change in AgI is larger (only NiTi comes close at 25.5 K), even though the pressure required is somewhat larger.

On page 2, paragraph 2, the authors say "that biaxial stresses... are realisable in thin films [11, 12]". This should clarify that *changes* of stress are being discussed, and comment on whether the stress produced by the ferroelectric substrates would be large enough to drive the MC effects.

Is this biaxial stress really required, or could uniaxial/isotropic pressure be used? This would have implications for experimentalists.

The basic physical idea about how stress produces entropy changes in FICs should be explained near the start of the paper.

It is misleading to write (Page 2, line 4) that MC strengths for ferroelectrics are < 1K/GPa, as ferroelectric ammonium sulphate shows 80 K/GPa (Table 1).

Line 4: "low-energy efficiency" in vapour compression is often claimed but incorrect.

Below fig 1: "as the size of SMA shrinks towards the nanoscale the martensitic phase transformation may be suppressed due to overstabilisation of the high-T distorted phase [5]."... the relevance of this for cooling is not immediately apparent, as a small SMA could not cool very much. Perhaps the authors are referring to sub-divided material in a fluid heat exchanger.

Overall, the prediction of large mechanocaloric effects near room temperature in a class of material that is well known, but not known for this purpose, represents an attractive development, and should inspire experimentalists to study MC effects in AgI. Therefore, if the above issues are addressed, the paper should be published in Nature Communications.

Reviewer #2:

Remarks to the Author:

This paper presents some interesting results of simulations of mechanocaloric effects. However, the manuscript is poorly written, has poor English and malapropisms ("The giant room-temperature MC effect unravelled in AgI thin films"), and the explanations even of what was done are impossible to follow. The supplementary material is too important and should be incorporated in the main text. The paper should be rewritten and submitted to a journal where full-length articles are appropriate. The paper says it is on thin-films of Li-rich (Li₃OCl) and type-I (AgI), but the supplemental material says CaF₂ and Li₃OCl. Why are "thin-films" important, as in the title? And how are the present results then relevant? The methods section says "Large simulation boxes, typically containing 6,144 atoms, are used in which periodic boundary conditions along the three Cartesian directions are applied." --so not thin films. And why do superionic materials have large mechanocaloric effects, if what makes them superionic, their defects, actually inhibit the mechanocaloric effect, according to this paper. Based on all of the above, this paper is not suitable for publication.

=====
Reviewer 1
=====

The paper reports predictions of mechanocaloric effects in ionic conductors (38 K for 1 GPa in AgI near room temperature, and 9 K for 1 GPa in Li3OCl at high temperature). The main result differs from the authors' previous work [9] because the effects in AgI occur near room temperature. Compared with other materials in Table 1, the predicted temperature change in AgI is larger (only NiTi comes close at 25.5 K), even though the pressure required is somewhat larger.

A: We would like to thank Reviewer 1 for his/her careful reading of our work and very useful suggestions and comments.

*On page 2, paragraph 2, the authors say “that biaxial stresses... are realisable in thin films [11, 12]”. This should clarify that *changes* of stress are being discussed, and comment on whether the stress produced by the ferroelectric substrates would be large enough to drive the MC effects.*

A: We agree with Reviewer 1 in that dynamic changes of biaxial strain (or, equivalently, of biaxial stress) is the key quantity for the practical realization of mechanocaloric effects in thin films. Achievement of dynamic changes of biaxial strain in thin films of the order of 0.1-1.0% have been reported in the literature, which involve the use of (i) elastic bending techniques [1], (ii) piezoelectric substrates [2], and (iii) “nanojigs”, that is, three-point beam bending techniques [3].

[1] Singh, S.; Fitzsimmons, M. R.; Lookman, T.; Jeen, H.; Biswas, A.; Roldan, M. A.; Varela, M., *Role of elastic bending stress on magnetism of a manganite thin film studied by polarized neutron reflectometry*, Phys. Rev. B 2012, **85**, 214440-6.

[2] Liu, Y.; Phillips, L. C.; Mattana, R.; Bibes, M.; Barthelemy, A. ; Dkhil, B., *Large reversible caloric effect in FeRh thin films via a dual-stimulus multicaloric cycle*. Nat. Comm. 2016, **7**, 11614.

[3] Tosado, J.; Dhakal, T.; Biswas, A., *Colossal piezoresistance in phase separated manganites*, J. Phys.: Condens. Matt. 2009, **21**, 192203-3.

Even larger biaxial strain changes (5%<) can be envisaged with the use of mechanical extensometer techniques applied on polymeric flexible substrates [4], and of nanoindentation methods [5].

[4] Tusek, J.; Engelbrecht, K.; Mikkelsen, L. P; Pryds, N., *Elastocaloric effect of Ni-Ti wire for application in a cooling device*. J. Appl. Phys. 2015, **117**, 124901.

[5] Pathak, S.; Kalidindi, S. R., *Spherical nanoindentation stress-strain curves*. Mater. Sci. Eng. R 2015, **91**, 1.

We should note that in the case of AgI thin films, we have predicted quite large dynamic changes of epitaxial strain of the order of 10%. However, even if just considering the case of ferroelectric substrates, which are able to inflict dynamic changes of epitaxial strain of the order of 0.1-1.0%, this should not be a problem for the implementation of likely solid-state cooling applications. Namely, the AgI samples could be always prepared, either by means of chemical doping (e.g, with Rb ions [5]) or by introducing an initial epitaxial strain in the system, close enough to the transition point that triggers the giant mechanocaloric effect, so that relatively small epitaxial strain changes could be sufficient to move forward and backwards on the transition.

[5] Hull, S., *Superionics: Crystal structures and conduction processes*. Rep. Prog. Phys. 2004, **67**, 1233.

Following the Reviewer 1's suggestion, we have modified the part of the Introduction in which we commented on the realization of biaxial stresses in thin films (i.e., third paragraph after the Abstract). That part now reads as it follows:

“Biaxial stresses, either compressive, $\sigma > 0$, or tensile, $\sigma < 0$, are realisable in thin films [15]; dynamic biaxial stresses leading to epitaxial strain changes of the order of 0.1-1% have been demonstrated in thin films by using ferroelectric substrates [16,17], and larger dynamic changes can be envisaged by means of mechanical extensometer techniques applied on flexible polymeric substrates [18] and of nanoindentation methods [19].”

Is this biaxial stress really required, or could uniaxial/isotropic pressure be used? This would have implications for experimentalists.

A: This is indeed a very good question. Based on our previous work on fluorite-structured FIC (Nano Letters **16**, 3124 -2016-), in which we analysed the effects of uniaxial, biaxial, and hydrostatic pressures on the corresponding caloric properties, we can say that not only biaxial but also uniaxial and hydrostatic (Phys. Rev. Lett. **113**, 235902 -2014-) may be used to drive the superionic transition in FIC (and hence mechanocaloric effects). However, in order to make a quantitative prediction on the actual potential of uniaxial and hydrostatic stresses in Li-rich and type-I FIC, additional simulations should have to be undertaken.

In view of the Reviewer 1's comment, we have added the following sentence to the Discussion section:

“Mechanical stresses other than biaxial (i.e., uniaxial and hydrostatic) also can be expected to produce similar mechanocaloric responses than reported here in FIC [11].”

The basic physical idea about how stress produces entropy changes in FICs should be explained near the start of the paper.

A: Following the Reviewer 1's advice, we have revised the Introduction section in our article. The basic physical idea about how mechanical stress produces entropy changes in FIC, now is introduced in the paragraph where we explain the recently predicted giant mechanocaloric effect in fluorite-structured FIC (i.e., second paragraph after the Abstract). In particular, we write:

“Recently, a giant MC effect has been predicted in fluorite-structured fast-ion conductors (FIC), typified by CaF_2 and PbF_2 , which is comparable in magnitude to the benchmark adiabatic temperature shifts measured in SMA [11]. The MC effect disclosed in FIC may be understood in terms of stress-driven changes in ionic diffusivity, which in turn cause sizeable lattice strains [11]. The originating superionic transition is of second-order type, in analogy to archetypal FE, and fluorite-structured FIC also can be synthesised as nanomaterials [12].”

It is misleading to write (Page 2, line 4) that MC strengths for ferroelectrics are $< 1\text{K/GPa}$, as ferrielectric ammonium sulphate shows 80K/GPa (Table 1).

A: Following the Reviewer 1's advice, we have revised the Introduction section in our article as regards the assessment of ferroelectric materials as potential mechanocaloric materials. In particular, now we write:

“Ferroelectric compounds (FE) like, for instance, BaTiO_3 [6], PbTiO_3 [7], and $\text{Ba}_{1-x}\text{Ca}_x\text{Ti}_{1-y}\text{ZrO}_3$ solid solutions [8] (typically exploited in sensing, information storage and energy applications), can be synthesised as nano-sized materials and generally exhibit displacive second-order phase transitions [9]. However, current adiabatic temperature shifts, ΔT , estimated **in most** FE materials near room temperature are about one order of magnitude smaller than those achieved in SMA **(made the exception of the ferrielectric compound $(\text{NH}_4)_2\text{SO}_4$ [10])**, and the involved mechanical stresses appear to be unsuitably too large ($\sigma \gg 1\text{GPa}$). Consequently, the MC strengths reported so far for FE are rather poor **in general**, namely, $|\Delta T / \sigma| < 1\text{K}\cdot\text{GPa}^{-1}$ **(again, made the exception of $(\text{NH}_4)_2\text{SO}_4$ [10])**.”

We should clarify that the promising mechanocaloric properties observed in the ferrielectric compound $(\text{NH}_4)_2\text{SO}_4$ in fact appear to be an exception as compared to the large number of ferroelectric materials (mostly oxide perovskites) which have been already investigated for caloric purposes. Our point here is to provide a general comparison between different families of well-established mechanocaloric materials.

Line 4: “low-energy efficiency” in vapour compression is often claimed but incorrect.

A: Following the Reviewer 1's advice, we have removed such a statement from the Introduction (see first paragraph after the Abstract).

Below fig 1: “as the size of SMA shrinks towards the nanoscale the martensitic phase transformation may be suppressed due to overstabilisation of the high-T distorted phase [5].”... the relevance of this for cooling is not immediately apparent, as a small SMA could not cool very much. Perhaps the authors are referring to subdivided material in a fluid heat exchanger.

A: Our point here is on the scalability problems encountered in SMA when moving down to the nano scale. We should agree with Reviewer 1 in that a single nanostructured mechanocaloric element may not be too useful when it comes to cooling. However, in the same way as assemblies of nano-elements (e.g., nanoparticles and nano-platelets) are used for gas storage, catalysis, and electrochemical applications operating in the macroscale scale, composites of nano-sized mechanocaloric elements could be exploited for solid-state refrigeration. In that case, the structural transformation should be preserved in each nano element in order to maximise the overall caloric response of the composite material.

Overall, the prediction of large mechanocaloric effects near room temperature in a class of material that is well known, but not known for this purpose, represents an attractive development, and should inspire experimentalists to study MC effects in AgI. Therefore, if the above issues are addressed, the paper should be published in Nature Communications.

A: We would like to thank Reviewer 1 for his/her positive recommendation on our work and very useful comments and suggestions. We hope that Reviewer 1 will find the revised version of our manuscript satisfactorily improved.

=====
Reviewer 2
=====

This paper presents some interesting results of simulations of mechanocaloric effects.

A: We would like to thank Reviewer 2 for his/her reading of our work.

However, the manuscript is poorly written, has poor English and malapropisms ("The giant room-temperature MC effect unravelled in AgI thin films"), and the explanations even of what was done are impossible to follow.

A: We may agree with Reviewer 2 in that the language in our article probably can be improved, as none of us is a native speaker of English. In the revised version of our article, we have tried to improve this aspect to the extent of our possibilities. In case the Editor considered it as necessary, we could always contract some professional editing service in order to further improve the language in our manuscript.

Nevertheless, we believe that, in contrast to Reviewer 2's opinion, the explanations that we provide in our article are sufficient to correctly understand what we have done and to follow our reasonings. We would like to mention here the case of Reviewer 1, for instance, who has provided a positive and physically very insightful assessment of our work.

The supplementary material is too important and should be incorporated in the main text. The paper should be rewritten and submitted to a journal where full-length articles are appropriate.

A: We believe that including too many technical details in the main sections of the article might go against one of our main interests, namely, to motivate experimental searches of the novel mechanocaloric phenomena that we report. On the other hand, we may agree with Referee 2 in that our paper could be expanded in order to improve its readability.

Following the Reviewer 2's recommendation, we have moved one figure from the Supplementary Material to the main text (see new Figure 6 therein). In the new figure we show the ionic radial pair distributions functions calculated for AgI in the zincblende structure as a function of biaxial stress at room temperature. The results shown in that figure were already discussed in the main text, however, in the revised version of our article we have changed slightly our explanations in order to provide a bit more of detail on them (see the two paragraphs preceding the Discussion section).

As regards the rest of information contained in the Supplementary Material, we firmly believe that it is not necessary to incorporate it in the main text. As a matter of fact short papers are more likely to be read and cited than long papers, and we consider that readers would be already able to follow and understand our explanations.

The paper says it is on thin-films of Li-rich (Li₃OCl) and type-I (AgI), but the supplemental material says CaF₂ and Li₃OCl. Why are “thin-films” important, as in the title? And how are the present results then relevant?

A: In our work, as it is indicated in the title, we report results on the mechanocaloric properties of superionic thin films. Apparently, Reviewer 2 is a bit confused because in the Supplementary Material we have presented a validation study of the employed force fields, in which we have carried out benchmark tests in the analogous bulk materials (as this is the most general system case to be considered, and it is customarily done in computational materials science).

Answering to the question of why thin films are important for mechanocaloric effects, essentially this is because either compressive ($\sigma > 0$) or tensile ($\sigma < 0$) stresses can be realised in practice by employing specific substrates and strain-stress methods (see third paragraph after the Abstract). In the case of barocaloric bulk materials, for instance, only compressive stresses ($\sigma > 0$) can be achieved. We note that in two of the three FIC thin films that we have analysed (i.e., CaF₂ and Li₃OCl) tensile stresses ($\sigma < 0$) are most favorable for the triggering of giant mechanocaloric effects; this means that similar caloric phenomena probably cannot be reproduced in the corresponding bulk systems by employing hydrostatic pressure. We note that uniaxial stresses also allow for application of compressive and tensile strain deformations on unidimensional materials; however, due to the high surface-to-volume ratio of thin films, faster heat transfer times and higher cycling frequencies are expected to be achieved with biaxial stresses on thin films [1].

[1] Ossmer, H., Lambrecht, F., Gultig, M., Chluba, C., Quandt, E. & Kohl, M. *Evolution of temperature profiles in TiNi films for elastocaloric cooling*. Acta Mater. **81**, 9 (2014).

The methods section says “Large simulation boxes, typically containing 6,144 atoms, are used in which periodic boundary conditions along the three Cartesian directions are applied.” --so not thin films.

A: Reviewer 2 appears to be misunderstanding something here. As a matter of fact, thin films are customarily simulated in computational studies by applying periodic boundary conditions along the three Cartesian directions. The mechanical condition that rigorously defines (fully strained) thin films is: $\sigma_{xx} = \sigma_{yy}$ and $\sigma_{zz} = 0$, and this

condition can be reproduced exactly in computer simulations by using barostatting techniques, when considering finite-temperature conditions, or by constraining the orientation and length of two (out of the three) lattice vectors, when considering $T=0$ conditions; in both cases, periodic boundary conditions are applied along the three Cartesian directions (see, for instance, works [2-4]).

[2] Lee, J. H. & Rabe, K. M. *Epitaxial-strain induced multiferroicity in SrMnO_3 from first-principles*. Phys. Rev. Lett. **104**, 207204 (2010).

[3] Pertsev, N. A., Zembilgotov, A. G. & Tagantsev, A. K. *Effect of mechanical boundary conditions on phase diagrams of epitaxial ferroelectric thin films*. Phys. Rev. Lett. **80**, 1988 (1998).

[4] Dieguez, O., Rabe, K. M. & Vanderbilt, D. *First-principles study of epitaxial strain in perovskites*. Phys. Rev. B **72**, 144101 (2005).

By applying fully periodic boundary conditions in our simulations, we do not need to explicitly simulate the substrate over which the thin film is grown (hence the simulations can be performed more efficiently); also, we are neglecting possible elastic relaxation effects and the interactions with vacuum at the top surface of the thin film. In other words, in our simulations we can single out the fundamental mechanocaloric effects that occur in FIC thin films. In practice, certainly, thin films are prone to suffer from elastic relaxation problems, which depend on their specific thickness, and vacuum-interface effects; however, those practical issues are extrinsic to the fundamental mechanocaloric phenomena that we are investigating here.

And why do superionic materials have large mechanocaloric effects, if what makes them superionic, their defects, actually inhibit the mechanocaloric effect, according to this paper.

A: Reviewer 2 also appears to be misunderstanding something here. Ionic vacancies, that is, stoichiometric point defects, are not always necessary for the triggering of superionicity. Actually, in two, out of the three, systems that we have analysed in this work (i.e., CaF_2 and AgI thin films) ionic conductivity is observed in the total absence of ionic vacancies (see Results section and references therein); we note that these results are well-established within the community of FIC researchers, both at the theoretical and experimental levels (see, for instance, work [5]). Furthermore, in the case of Li_3OCl thin films we have found that tensile stresses alone can actually induce ionic conductivity in the perfect stoichiometric system (this and other results on the effects of biaxial stress on the ionic conductivity properties of FIC thin films, will be published somewhere else in the near future). Therefore, ionic vacancies alone is not what makes FIC superionic (or, at least, not in the cases analysed in this work).

[5] Hull, S., *Superionics: Crystal structures and conduction processes*. Rep. Prog. Phys. 2004, **67**, 1233.

We would like to comment here on the differences between ionic vacancies and Frenkel pair defects, which are both different types of point defects. Ionic vacancies are stoichiometric point defects that involve complete removal of atoms from the system; on the other hand, Frenkel pair defects involve the simultaneous creation of ionic vacancies and interstitials (that is, there is not actual removal of atoms from the system), as due to the hop of ions through favorable pathways within the corresponding structural frameworks. In general, the creation of Frenkel pair defects is necessary for the triggering of ionic conductivity in FIC, whereas the creation of ionic vacancies is not. On the other hand, as we explain in the Results section, ionic vacancies can actually enhance the ionic conductivity of FIC as due to an effective increase of the space available to interstitial ions.

One thing that we have done in this work is to analyse, for the first time, the effects of ionic vacancies on the mechanocaloric performance of cooling FIC materials. In order to avoid possible misunderstandings, in the revised version of our manuscript we have specified that the class of point defects that we have considered in that part of our analysis corresponds to ionic vacancies, rather than to any other type of point defects (see Abstract and the Results section).

Based on all of the above, this paper is not suitable for publication.

A: Our impression is that Reviewer 2 is slightly skeptical on our results, essentially because he/she gets a bit confused with the technical details of our calculations. We have responded to each of his/her criticisms in a well reasoned manner, and when possible we have amended our manuscript accordingly. Therefore, we invite Reviewer 2 to take our responses into consideration with an open attitude.

Reviewers' Comments:

Reviewer #1:

Remarks to the Author:

The responses and changes are good, and the paper should be published if one outstanding problem is fixed: the statement about the basic mechanism seems incorrect. It says:

"The MC effect disclosed in FIC may be understood in terms of stress-driven changes in ionic diffusivity, which in turn cause sizeable lattice strains"...

... but the MC effect is a thermal change, and the statement does not explain how thermal changes come about. I would like to see the revision before finally recommending publication.

Reviewer #2:

Remarks to the Author:

The scientific responses from the authors make clear all of my scientific questions about their paper, and make clear that the work itself is well deserving of publication. Unfortunately, the paper itself is not in clarity. Nowhere do I see any explanation about how, for example, the computations present are related to thin films. Again, the response is very clear, but the paper itself is not. If the paper is only meant for expert theorist material scientists, this is not the right journal for it.

Reviewer #1 (Remarks to the Author):

The responses and changes are good, and the paper should be published if one outstanding problem is fixed: the statement about the basic mechanism seems incorrect. It says:

“The MC effect disclosed in FIC may be understood in terms of stress-driven changes in ionic diffusivity, which in turn cause sizeable lattice strains”...

... but the MC effect is a thermal change, and the statement does not explain how thermal changes come about. I would like to see the revision before finally recommending publication.

A: We would like to thank Reviewer #1 for his/her careful analysis of our work and positive recommendation.

Answering to the point raised by Reviewer #1, the predicted MC effect in FIC in fact is related to the large entropy change associated to the superionic phase transition, which typically is of the order of $\sim 100 \text{ JK}^{-1}\text{kg}^{-1}$; such a large change of entropy actually is the responsible for the large thermal changes observed under adiabatic conditions. Simultaneously, FIC undergo sizeable lattice strains during the normal to superionic phase transition.

In order to avoid possible misunderstandings, we have modified the sentence noted by Reviewer #1 as it follows: “The MC effect disclosed in FIC may be understood in terms of stress-driven changes in ionic diffusivity, which in turn cause large variations in the entropy and dimensions of the material.”

Reviewer #2 (Remarks to the Author):

The scientific responses from the authors make clear all of my scientific questions about their paper, and make clear that the work itself is well deserving of publication. Unfortunately, the paper itself is not in clarity. Nowhere do I see any explanation about how, for example, the computations present are related to thin films. Again, the response is very clear, but the paper itself is not. If the paper is only meant for expert theorist material scientists, this is not the right journal for it.

A: We would like to thank Reviewer #2 for his/her analysis of our work.

In order to avoid any possible misunderstanding on the connection between our calculations and thin films, as suggested by Reviewer #2's comments, we have added the following explanatory sentences in the Methods section:

“We note that by using periodic boundary conditions in our calculations we avoid to explicitly simulate the substrate over which the thin film is grown in practice. Also, possible elastic relaxation effects and the interactions of the thin film with the vacuum at the top surface are totally neglected. Consequently, the simulations are performed very efficiently in terms of computational expense and the fundamental mechanocaloric effects occurring in FIC thin films can be singled out.”